**Data Availability Statement:** Data cannot be shared publicly because they belong to the National Health Insurance Service (NHIS). There are ethical

# Trends in the incidence and prevalence of dysphagia requiring medical attention among adults in South Korea, 2006–2016: A nationwide population study

SuYeon Kwon[1], Seungwoo Cha[2], Junsik Kim[2], Kyungdo Han[3], Nam-Jong Paik[2], Won-Seok Kim📷[2]*

1 Department of Rehabilitation Medicine, Ewha Woman's University Seoul Hospital, School of Medicine, Seoul, Republic of Korea, 2 Department of Rehabilitation Medicine, Seoul National University College of Medicine, Seoul National University Bundang Hospital, Seongnam-si, Gyeonggi-do, Republic of Korea, 3 Department of Statistics and Actuarial Science, Soongsil University, Seoul, Republic of Korea

* wondol77@gmail.com

## Abstract

The prevalence of dysphagia is increasing, resulting in socioeconomic burden, but previous reports have only been based on a limited populations. Therefore, we aimed to investigate the nationwide incidence and prevalence of dysphagia requiring medical attention to provide adequate information for healthcare planning and resource allocation. In this nationwide retrospective cohort study, the data of adults aged ≥20 years recorded from 2006 to 2016 were sourced from the Korean National Health Insurance Service database. Medical claim codes based on ICD-10-CM were used to define dysphagia and possible causes. The annual incidence and prevalence of dysphagia were calculated. Cox regression was used to estimate dysphagia risk in people with possible dysphagia etiology. Survival analysis was performed to estimate the mortality and hazard ratio of dysphagia. The crude annual incidence of dysphagia increased continuously from 7.14 in 2006 to 15.64 in 2016. The crude annual prevalence of dysphagia in 2006 was 0.09% and increased annually to 0.25% in 2016. Stroke (odds ratio [OR]: 7.86, 95% confidence interval [CI]: 5.76–6.68), neurodegenerative disease (OR: 6.20, 95% CI: 5.76–6.68), cancer (OR: 5.59, 95% CI: 5.17–6.06), and chronic obstructive pulmonary disease (OR: 2.94, 95% CI: 2.71–3.18) were associated with a high risk of dysphagia. The mortality in the dysphagia group was 3.12 times higher than that in the non-dysphagia group (hazard ratio: 3.12, 95% CI: 3.03–3.23). The incidence and prevalence of dysphagia requiring medical attention are increasing annually. The increasing trend was conspicuous in the geriatric population. The presence of stroke, neurodegenerative disease, cancer, and chronic obstructive pulmonary disease is associated with a high risk of dysphagia. Therefore, adequate screening, diagnosis, and management of dysphagia in the older population must be emphasized in geriatric healthcare.

restrictions on sharing a data set because data contain potentially identifying or sensitive patient information.To request data from NHIS(access number NHIS-2018-1-331), researchers have to apply during the recruitment period and submit a research proposal. The committee reviews the proposals and then selects a few researchers to use and analyze the data. Data access applications for the national health insurance data are available on the NHIS data sharing website (https://nhiss. nhis.or.kr/bd/ab/bdaba000eng.do). Raw data was available to the researchers upon reasonable academic request and with the permission of the Korean NHIS Institutional Data Access (http:// nhiss.nhis.or.kr). The authors had no special access privileges.

**Funding:** This work was supported by Seoul National University Bundang Hospital (https:// www.snubh.org) Research Fund (No. 06-2018-140). The funders did not play a role in the study design, data collection and analysis, decision to publish, or preparation of the manuscript.

**Competing interests:** The authors have declared that no competing interests exist.

## Introduction

Swallowing is the process of passing solid and liquid food from the oral cavity, pharynx, and esophagus into the stomach with appropriate coordination [1]. Normal physiological swallowing is important not only for adequate nutrition and hydration but also for prevention of complications, such as aspiration pneumonia and asphyxia [2, 3]. The World Health Organization has defined dysphagia as the difficulty in forming or moving a bolus safely from the oral cavity to the esophagus [4], and has recognized it as a medical disability associated with increased morbidity, mortality, and cost of treatment [1, 3].

The swallowing process can be disrupted by diverse diseases such as stroke, traumatic brain injury, neurodegenerative disease (ND), motor neuron diseases, or cancers [5, 6]. Since these diseases are increasing with an aging of the population, the incidence of dysphagia, as one of the associated complications, is expected to increase accordingly [7, 8]. In addition, aging is associated with dysphagia due to muscle wasting, delayed swallowing reflex, and neuromuscular incoordination [2, 3, 9, 10]. Dysphagia is becoming more common in the geriatric population [8] as life expectancy increases. The prevalence of dysphagia in the geriatric population has been reported variably in different countries. In the United States, the prevalence of dysphagia in adults over 50 years of age ranges from 11.4% to 22% [5, 11], and that in the geriatric population living in the community is 33% [12]. In Europe, the prevalence of dysphagia in the geriatric population living in the community ranges from 30% to 40%, and that in hospital/institutional settings ranges from 25.5% to 60% [4, 7]. In Asia, the prevalence of dysphagia in the geriatric population living in the community ranges from 13.8% to 33.7% [13, 14]. A meta-analysis and systematic review of the multinational studies reported that dysphagia affects 30% of community-dwelling geriatric population, almost 50% of geriatric patients, and above 50% of nursing home residents [15]. Therefore, diagnosing and managing dysphagia is an important issue in geriatric healthcare.

The previously reported prevalence of dysphagia is not representative of the entire population because most of these reports have come from cross-sectional studies with small sample sizes, specific diseases (e.g., stroke, ND, cancer) [16–20], and specific healthcare settings (hospitalized/institution settings) with specific age groups (healthy or frail geriatric population) [12–14]. Hence, there is a lack of nationwide data on the prevalence, incidence, and mortality of dysphagia to establish relevant healthcare policies on the evaluation and management of dysphagia.

Therefore, the primary purpose of this study was to investigate the nationwide trend of the incidence and prevalence of dysphagia requiring medical attention from a large, representative, population-based cohort, using claim data from the Korean National Health Insurance Service (NHIS) database from 2006 to 2016. We further investigated the possible diseases associated with dysphagia, and the risk of long-term mortality in dysphagic individuals compared to non-dysphagic individuals.

## Materials and methods

### Data source and study population

The data source for this study is the NHIS database of South Korea. The NHIS has granted access to researchers since 2014. The NHIS is a universal social health insurance that provides healthcare coverage and medical aid to most of the Korean population (97.2% and 2.8%, respectively) [21]. Therefore, the NHIS presumptively has all records of medical service claims and sociodemographic information of the entire Korean population. The claim codes used in the Korean NHIS are based on the International Classification of Disease-Tenth Revision,

Clinical Modification (ICD-10-CM). The complete data of the adult population aged ≥20 years (inpatient and outpatient medical service claims and sociodemographic information) from 2006 to 2016 were extracted. This study was exempt from review by the Seoul National University Bundang Hospital Institutional Review Board (X-1807/483-904), complying with the requirements of the Declaration of Helsinki. Data from the Korean NHIS were fully anonymized for analyses, and the need for obtaining informed consent was waived.

### Definition of dysphagia and diseases that may have caused dysphagia

To define dysphagia, diagnostic codes and medical claim codes related to diagnosis or treatment of dysphagia were used. If a person matched at least one of the following criteria, the individual was assumed to have dysphagia: 1) presence of the diagnostic code for dysphagia (R13), 2) swallowing therapies (MX141) at least twice within 1 month, 3) instrumental swallowing tests (E7011, E7012) such as videofluoroscopic swallowing studies (VFSS) or fiberoptic endoscopic evaluation of swallowing (FEES) at least twice within 3 months, 4) nasogastric tube insertions (Q2621, Q2622) at least twice within 3 months, or 5) procedure code claims for percutaneous gastrostomy (M6730, Q2612) at least once. In summary, the operational definition tried to include dysphagic people who received medical attention for dysphagia diagnosis or management during admission or outpatient visits.

Medical claim codes based on ICD-10-CM were used to define the diseases that may have caused dysphagia: 1) stroke as diagnostic codes of I63 and I64 and brain imaging (computed tomography or magnetic resonance imaging) during admission, 2) ND as diagnostic codes claimed for Parkinson's disease (G20 and V124) or those claimed for dementia (F00-03, G30, G31, G231, G310, G318, and F107), 3) cancer as diagnostic code of C codes (C00-14, C15-25, C32-34, C50, C53, C54, C56, C61, C62, C64, C67, C70-72, C73, C81-86, C90-95) and V193, and 4) chronic obstructive pulmonary disease (COPD) as diagnostic codes of J41-J44. V193 and V194 are specific deductible insurance codes for cancer and Parkinson's disease respectively in Korea. These four diseases were selected because they have been associated with dysphagia in previous studies [18, 19, 22].

### Other variables

Other epidemiologic factors including age, sex, and income, were obtained and stratified into quartiles for statistical analyses. Data regarding death from various causes were obtained from the NHIS database linked to Statistics Korea, which records deaths of Korean citizens.

### Statistics

The crude annual incidence was calculated by dividing the number of new patients with dysphagia each year by 10,000 person-years at risk. People diagnosed with dysphagia before 2006 were excluded as they would not represent new cases. The age-adjusted incidence and the incidence of dysphagia in each age group were calculated for each year to control for different age distributions throughout the years. The crude annual prevalence was calculated by dividing the number of people with dysphagia each year by the total of 10,000 people. Individuals who did not meet the operational definition of dysphagia in that year were excluded, even though they might have met the definition in the previous years. The age-adjusted prevalence and prevalence of dysphagia in each age group were also calculated.

To investigate the risk of dysphagia in possible common etiologies of dysphagia and the risk of mortality in dysphagia, data of people who were defined as having dysphagia for the first time in 2010 (n = 39795) were extracted and these individuals were followed up until 2016; these data were compared with the data of the age- and sex-matched population without

dysphagia (n = 39795). The chi-square test was used to compare categorical variables between the dysphagia and non-dysphagia groups. Logistic regression analysis was used to determine the risk of dysphagia in each possible common etiology of dysphagia (stroke, ND, cancer, and COPD) in 2010. Adjustments for age, sex, low-income status, and comorbidities were made. Multivariate adjusted Cox proportional hazard regression analysis was used to identify the risk of mortality in the dysphagia group. Time-to-event analysis was used to estimate the mortality rate, and a Kaplan–Meier survival curve was plotted and compared using the log-rank test. All statistical analyses were performed using SAS version 9.4 (SAS Institute Inc., Cary, NC, USA). Statistical significance was set at P < 0.05.

## Results

### Incidence of dysphagia

The crude annual incidence of dysphagia increased every year from 7.14 in 2006 to 15.64 in 2016, and the age-adjusted annual incidence showed a similar increasing trend (**Fig 1A–1C**). In addition, the annual incidence of dysphagia increased more prominently in the geriatric group aged >70 years. (**Fig 1D–1F**).

### Prevalence of dysphagia

The crude annual prevalence of dysphagia showed a 2.8-fold increase over the 10 years, and the age-adjusted prevalence showed a similar increasing trend (**Fig 2A–2C**). In addition, the increasing trend of prevalence was steeper in the geriatric group aged >70 years (**Fig 2D–2F**).

### Risk of dysphagia by possible underlying etiologies

In the sample for the cross-sectional analysis (n = 39795 in each group), there were no differences in age or sex between the dysphagia and non-dysphagia groups due to age and sex

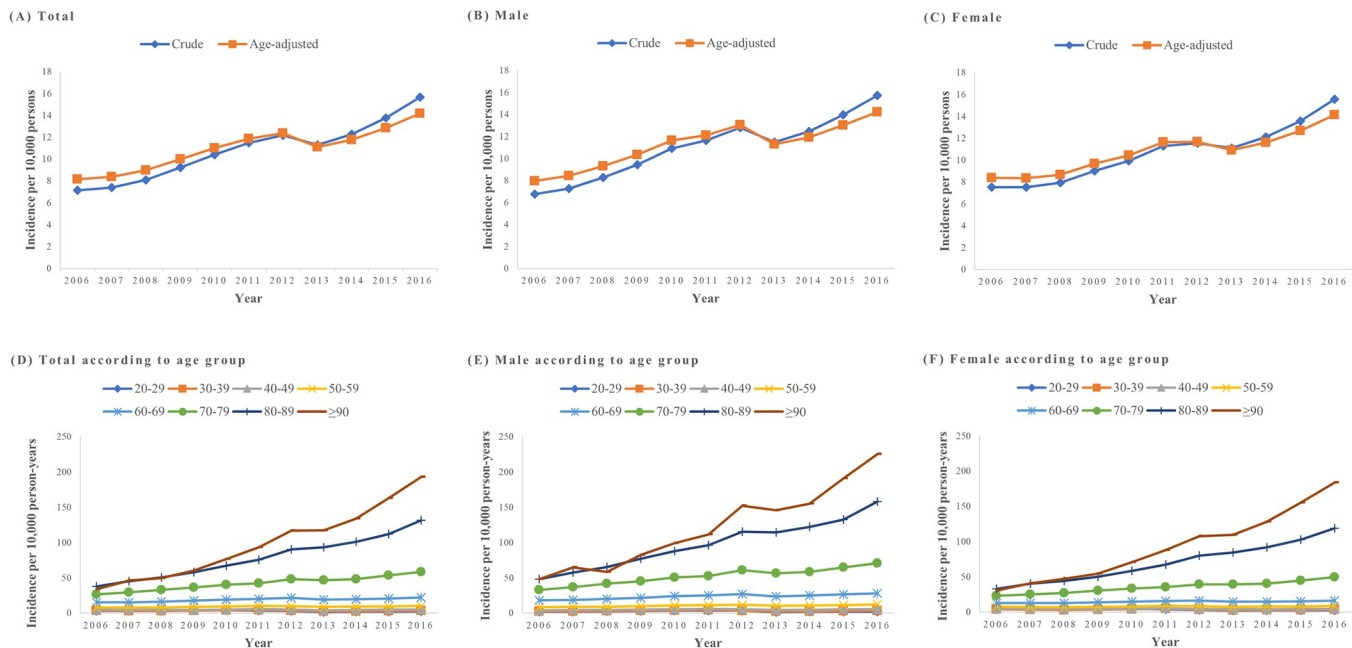

**Fig 1.** Incidence of dysphagia between 2006 and 2016 (A) total, (B) male, (C) female, (D) total according to age group, (E) male according to age group, (F) female according to age group.

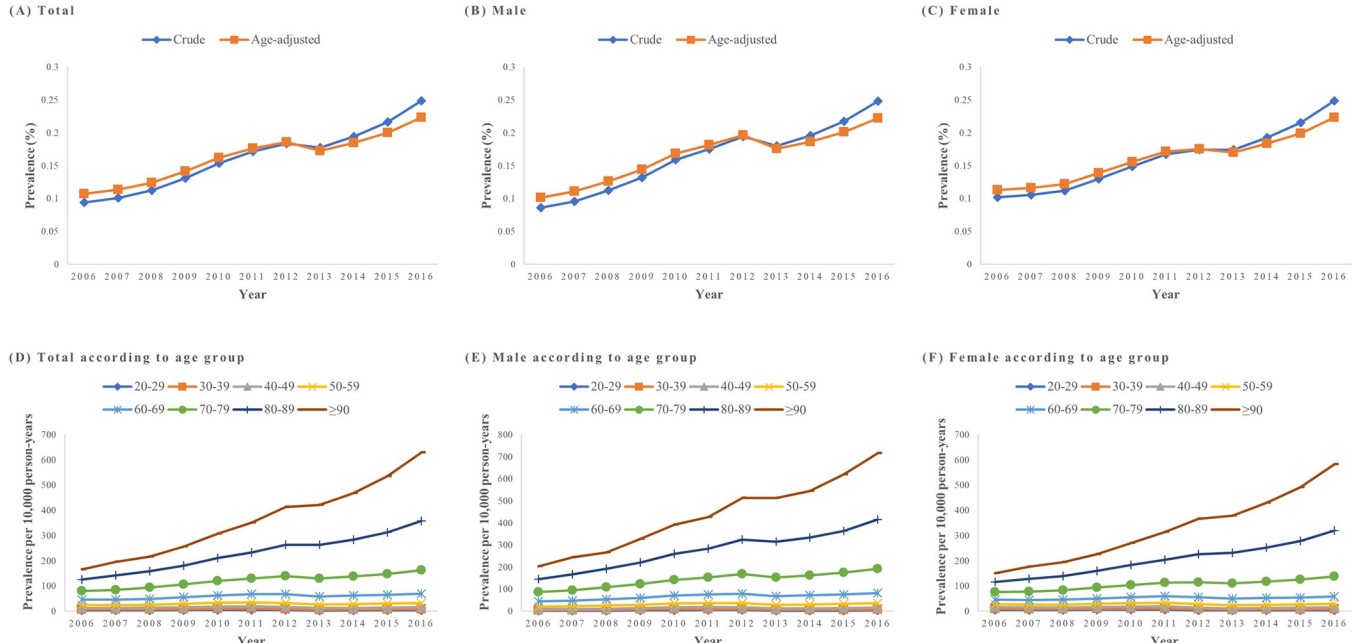

**Fig 2.** Prevalence of dysphagia (A) total, (B) male, (C) female, (D) total according to age group, (E) male according to age group, (F) female according to age group.

matching; however, a large number of individuals with dysphagia had low income and comorbidities compared to individuals without dysphagia (**S1 Table**). The risk of dysphagia was significantly higher in each common etiology of dysphagia, with adjustments for age, sex, low-income status, and comorbidities (**Table 1**).

The risk of dysphagia among the sampled comorbidities is noted from greatest to less association as follows: stroke (odds ratio [OR]: 7.86, 95% confidence interval [CI]: 5.76–6.68);

**Table 1. Odds ratio of dysphagia by possible etiologies.**

|  | Number of total individuals | Number of individuals with dysphagia | % of dysphagia | Odds ratio (95% CI) | |
|---|---|---|---|---|---|
|  |  |  |  | Crude | Model 1* |
| **Stroke** |  |  |  |  |  |
| Yes | 23604 | 19074 | 80.8 | 9.54 (9.07–10.02) | 7.86 (5.76–6.68) |
| No | 55986 | 20721 | 37.0 | 1 (reference) | 1 (reference) |
| **ND** |  |  |  |  |  |
| Yes | 13814 | 11590 | 83.9 | 10.00 (9.38–10.66) | 6.20 (5.76–6.68) |
| No | 65776 | 28205 | 42.9 | 1 (reference) | 1 (reference) |
| **Cancer** |  |  |  |  |  |
| Yes | 6373 | 4828 | 75.8 | 3.54 (3.33–3.76) | 5.59 (5.17–6.06) |
| No | 73217 | 34967 | 47.8 | 1 (reference) | 1 (reference) |
| **COPD** |  |  |  |  |  |
| Yes | 7028 | 5179 | 73.7 | 3.25 (3.06–3.44) | 2.94 (2.71–3.18) |
| No | 72562 | 34616 | 47.7 | 1 (reference) | 1 (reference) |

All *p*-values were < 0.001.

*A logistic regression analysis adjusted for age, sex, low-income status, and comorbidities.

CI: Confidence interval, ND: Neurodegenerative disease, COPD: Chronic obstructive pulmonary disease

p < 0.001), ND (OR: 6.20, 95% CI: 5.76–6.68; p < 0.001), cancer (OR: 5.59, 95% CI: 5.17–6.06; p < 0.001), and COPD (OR: 2.94, 95% CI: 2.71–3.18; p < 0.001).

## Mortality

The Kaplan-Meier plot presented in Fig 3 demonstrates a positive correlation between dysphagia and the risk of mortality. In addition, it is shown that this mortality risk increases yearly, particularly after the first 2 years of diagnosis (**Fig 3**).

The mortality rate in the dysphagia group was 93.11 per 1000 person-years, and the risk of death due to dysphagia increased 3.12 times (hazard ratio: 3.12, 95% CI: 3.03–3.23) even when adjusted for age, sex, low-income status, and comorbidities (**Table 2**).

## Discussion

In this study, it is shown that the overall incidence and prevalence of dysphagia increased with age. The increasing trend was conspicuous in the geriatric population, particularly those older than 70 years. To the best of our knowledge, this is the first report on the incidence, prevalence, and increasing trend of dysphagia in the entire adult Korean population. Most of the previously reported prevalence values of dysphagia came from questionnaire analyses or cross-sectional studies of small sample sizes, specific diseases, and specific populations [12–14, 16–20]. The relationship between dysphagia and aging can first be explained by geriatric syndromes and comorbidities [7, 8]. The swallowing function may decline with aging, followed by changes in swallowing physiology, such as weakness of the swallowing reflex, cough response, bolus propulsion, and increased residue [2, 14, 23–25]. The associated sarcopenia, frailty, pain, inadequate dentition, and polypharmacy may even aggravate the decline in swallowing

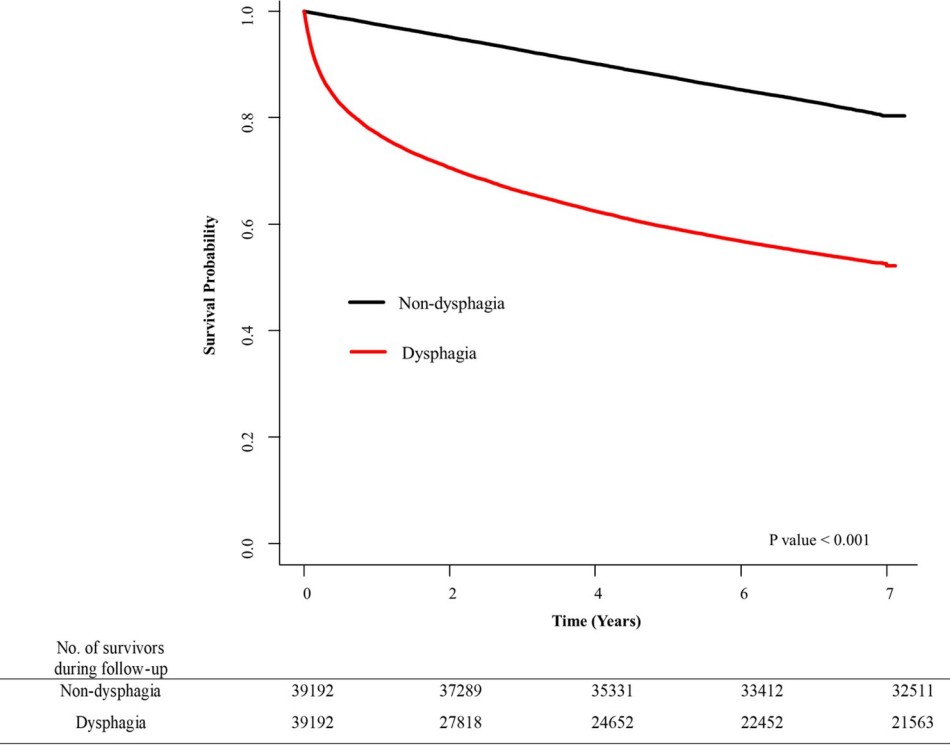

**Fig 3. Kaplan–Meier curve for mortality according to dysphagia.**

**Table 2. Hazard ratio for mortality by dysphagia.**

| Dysphagia | Number of individuals | Number of deaths | Duration (person-years) | Mortality rate (per 1000 person-years) | Hazard ratio (95% CI) | |
|---|---|---|---|---|---|---|
| | | | | | Crude | Model 1* |
| Yes | 39192 | 18037 | 193721.19 | 93.11 | 3.25 (3.16–3.34) | 3.12 (3.03–3.23) |
| No | 39192 | 7192 | 266974.32 | 26.94 | 1 (reference) | 1 (reference) |

All *p*-values were < 0.001.

*A Cox proportional hazard regression analysis adjusted for age, sex, low income status, and comorbidities.

CI: Confidence interval

function [7–9, 26, 27]. Polypharmacy in particular includes sedative drugs, opioids, and also higher anticholinergic burden, which may impair cognitive and swallowing function [4]. Second, this finding further suggests that the life expectancy of the Korean population has been extended such that a sufficient number of older individuals have been registered for the investigation and follow-up of the incidence and prevalence of dysphagia in this study. The proportion of those aged over 65 years in Korea was 14% in 2017 and is expected to reach 25% by 2030 [28]. As aging is one of the causes of dysphagia, dysphagia in the geriatric population will increase further and will be a more common geriatric disorder that requires medical attention. Lastly, this finding implies that dysphagia may place a significant socioeconomic and medical burden on the older population. The Korean national statistics reported that medical costs for those older than 65 years were about 40% of all medical costs in 2019 and that cardiovascular disease, neoplasm, respiratory disease, and nervous system disorders were the main causes of medical expenditures [29]. The current study demonstrates that stroke, cancer, COPD, and ND significantly increase the risk of dysphagia and that the presence of dysphagia significantly increases mortality. Therefore, dysphagia in older individuals should also be considered an important cause of socioeconomic and medical burden, demanding more substantial resources and adequate screening, diagnosis, and management independently of the underlying comorbidities, because in older adults the most prevalent type of dysphagia is oropharyngeal cause. Additionally, dysphagia burden may be highlighted because this analysis was conducted using the NHIS claim codes, including objectively defined dysphagia and etiologies, which were related to medical expenditure.

As identified in the cross-sectional analysis in this study, the presence of stroke, ND, cancer, and COPD was strongly associated with a high risk of dysphagia. This finding is consistent with that of previous studies [12, 16–20], and the pathophysiology of dysphagia in each disease can be explained as follows. A more severe stroke would initially have more severe neurological deficits, possibly including dysphagia [30, 31]. The lesion locations are also related to swallowing dysfunction. Cerebellar and brain stem lesions may impair swallowing physiology, and cerebral lesions may impair the oral phase, including mastication and bolus transport. Cortical lesions may impair orofacial motor control and pharyngeal peristalsis [16]. Moreover, cognitive deficits may impair the control of swallowing [16, 32]. Although the pathophysiology of dysphagia in Parkinson's disease is not clearly identified [33], it is believed that striatal dopaminergic deficiency and Lewy bodies affect swallowing centers and that advanced stages and prolonged disease duration are associated with severe dysphagia [34]. In dementia, the mechanisms of dysphagia depend on the type and progression of dementia. Given the nature of dementia, the impairment of food recognition, swallowing control, and cough response due to cognitive deficits may be related to the aggravation of dysphagia [17]. In cancers, dysphagia is related to the tumor site, advanced stage, and treatment modalities. Cancers, particularly of the head and neck and upper gastrointestinal tract, may interfere with food passage. Treatment

modalities such as tumor resection, chemotherapy, and radiotherapy may cause anatomical or neurological damage to swallowing function [35–37]. The increasing number of geriatric cancer survivors [38] is also assumed to increase the number of dysphagia cases. In COPD, respiratory-swallowing discoordination, tachypnea and/or dyspnea, oropharyngeal swallowing dysfunction, and physical and emotional distress are associated with an increased risk of dysphagia [12, 22, 39, 40]. In addition, this study simultaneously analyzed the relationship between dysphagia and various etiologies, including diseases and age factors, differentiating it from previous studies. Considering that dysphagia risk increases with comorbidity and that the increase in dysphagia is prominent in the geriatric population, it additionally indicates that the geriatric population are inevitably more susceptible to comorbidities and are unable to compensate for disease-related swallowing dysfunction [41].

The current study showed a relatively low prevalence of dysphagia compared to published studies. The crude prevalence of dysphagia ranged from 0.09% to 0.25%. Previous studies reported that the prevalence of dysphagia ranged from 16% to 22% in people older than 50 years [5, 11], 11.4% to 40% in geriatric population older than 65 years of age living in the community, and 25.5% to 60% in hospital/ institutional settings [4, 5, 7, 11–14]. This discrepancy is probably due to our operational definition, which included patients with dysphagia who received medical attention such as outpatient or inpatient physician care, instrumental swallowing tests, and swallowing therapies. Since the incidence and prevalence reported in this study are based on the records of claims, the current study excluded people with dysphagia who have not received medical services. Although physician care, instrumental swallowing tests, and swallowing therapies help evaluate dysphagia accurately, they have disadvantages of cost and time compared to self-reports or standardized swallowing assessments [9]. In addition, people with mild or transient dysphagia may have recovered without seeing a doctor or receiving medical care, thus could not be included in the dysphagia group. Consequently, the relatively low prevalence of dysphagia in the current study indicates more potentially unmet medical services than expected in the population with dysphagia. In addition, the lower prevalence of dysphagia in our study compared to that in previous studies is because this study was conducted in the general population of Korea. It is known that the prevalence of dysphagia is higher in hospital/institution settings than in community dwellings [42, 43].

The current study shows that dysphagia significantly increases long-term mortality. This finding is similar to the previous literature, which suggested that mortality due to dysphagia was associated with aspiration or asphyxia, regardless of the underlying cause [2, 32, 44]. Dysphagia fatality is additionally supported by previous literature, which reported that dysphagia could cause numerous complications, including malnutrition and dehydration, having a great impact on morbimortality, re-hospitalization, frailty, and quality of life [7, 17, 22, 30, 32, 34, 45]. As dysphagia can place a significant socioeconomic and medical burden on the Korean population, a careful analysis of the medical background is necessary to properly evaluate and treat dysphagia requiring medical attention, including the review of potentially causative diseases and medications.

This study has several strengths and limitations. First, the absolute numerical results in this study should be interpreted with caution because of the specificity and sensitivity of the operational definition. To minimize over- or underestimation, we tried to make an objective operational definition of dysphagia using the codes from the NHIS database and to specify dysphagia, which incurs medical expenses using claim codes of instrumental swallowing tests, swallowing therapies, and percutaneous gastrostomies. VFSS is considered the gold standard for identifying dysphagia [46, 47] and is routinely performed in South Korea. Therefore, we intended to increase the sensitivity and specificity of detecting dysphagia by including people who received VFSS or FEES at least twice within 3 months. When a penetration or aspiration

is detected by an instrumental swallowing test, the follow-up tests is usually performed within 3 months to examine the recovery or deterioration of swallowing [48, 49]. Conversely, people with preserved swallowing function are not followed up unless a sudden worsening of dysphagia occurs. We also included people who received multiple swallowing therapies within 1 month. If penetration and/or aspiration are detected by the instrumental swallowing tests, most people receive swallowing therapies at least once a week. In addition, the criterion of nasogastric tube insertions more than two times was used to define dysphagia, as similarly described in a previous study conducted in Taiwan [50]. People who underwent the percutaneous gastrostomy were included because most of the procedures are performed in patients with dysphagia [51]. This strict definition of dysphagia can lead to an underestimation of dysphagia due to the exclusion of subjective dysphagia and people with dysphagia but limited access to dysphagia evaluation and treatment. The possibility that dysphagia requiring medical attention was not captured due to limited access to medical services warrants further study to investigate this gap. However, this study is valuable because we could define objective dysphagia requiring medical attention, which is more directly associated with medical costs. Moreover, the current study used a long-term and large database to study the prevalence, incidence, and mortality of dysphagia at the general Korean population level. Although not a novel result, the current study provided robust evidence that the prevalence and incidence of dysphagia increase with age, aging and/or comorbidities multiply the risk of dysphagia, and the presence of dysphagia significantly increases mortality.

Second, a few factors might undermine the representativeness of the current study. The operational definition possibly included more oropharyngeal than esophageal dysphagia because the instrumental swallowing tests (VFSS and FEES) and the therapies were usually aimed at evaluating and improving oropharyngeal dysphagia. In addition, the exclusion of the pediatric population and related congenital diseases might have influenced the analysis of the prevalence, incidence, and mortality of dysphagia in the current study.

Third, the types and severity of dysphagia and their relationship with aging and comorbidities were not reflected. Since people who received instrumental swallowing tests and/or percutaneous gastrostomy would already have a high risk of dysphagia and more severe deficits [52], it is likely that apparent but severe dysphagia would be registered to satisfy the operational definition of dysphagia.

Fourth, only four diseases were selected as medical conditions that could be related to dysphagia, and the causal relationship between those possible etiologies and dysphagia could not be evaluated by cross-sectional analysis. Future studies should consider more diverse diseases, such as pneumonia and frailty, as possible etiologies of dysphagia.

Lastly, it is difficult to extrapolate the results from this study to the entire world because medical and epidemiologic circumstances of dysphagia may be different in various countries. Nevertheless, it is assumed that similar trends of incidence, prevalence, and mortality of dysphagia may be seen in the aging population in other countries where similar evaluation and treatment of dysphagia are being performed. In this regard, future studies on dysphagia using global data from various countries is necessary.

## Conclusion

The incidence and prevalence of dysphagia cases requiring medical attention are increasing yearly, and dysphagia increases long-term mortality. The presence of stroke, ND, cancer, and COPD is associated with a high risk of dysphagia. As many countries are entering an aging or super-aging society, dysphagia will be a more common problem that requires appropriate interventions in daily clinical practice such as the rest of geriatrics syndromes.

## Supporting information

**S1 Table. Baseline characteristics according to dysphagia in the year of 2010.**
(DOCX)

## Acknowledgments

The authors would like to thank the NHIS for the cooperation.

## Author Contributions

**Conceptualization:** SuYeon Kwon, Nam-Jong Paik, Won-Seok Kim.

**Data curation:** SuYeon Kwon, Kyungdo Han.

**Formal analysis:** Seungwoo Cha, Junsik Kim, Kyungdo Han.

**Funding acquisition:** Won-Seok Kim.

**Investigation:** SuYeon Kwon, Seungwoo Cha, Junsik Kim, Kyungdo Han, Won-Seok Kim.

**Methodology:** Kyungdo Han.

**Project administration:** SuYeon Kwon, Nam-Jong Paik, Won-Seok Kim.

**Resources:** SuYeon Kwon, Junsik Kim, Won-Seok Kim.

**Software:** Kyungdo Han.

**Supervision:** Nam-Jong Paik, Won-Seok Kim.

**Validation:** SuYeon Kwon, Seungwoo Cha, Won-Seok Kim.

**Visualization:** SuYeon Kwon, Junsik Kim.

**Writing – original draft:** SuYeon Kwon, Seungwoo Cha, Won-Seok Kim.

**Writing – review & editing:** SuYeon Kwon, Seungwoo Cha, Junsik Kim, Kyungdo Han, Nam-Jong Paik, Won-Seok Kim.

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
