## [Decision Letter · Decision Letter 0]

17 Jan 2023

PONE-D-22-35490Trends in the incidence and prevalence of dysphagia requiring medical attention among adults in South Korea, 2006–2016: A nationwide population studyPLOS ONE

Dear Dr. Kim,

Thank you for submitting your manuscript to PLOS ONE. After careful consideration, we feel that it has merit but does not fully meet PLOS ONE’s publication criteria as it currently stands. Therefore, we invite you to submit a revised version of the manuscript that addresses the points raised during the review process. The team of reviewers provided insightful comments on your submission. Please follow the comments carefully to address them in order to revise the submission for further decision.

We look forward to receiving your revised manuscript.

Kind regards,

Sina Azadnajafabad

Academic Editor

PLOS ONE

Journal Requirements:

"This work was supported by Seoul National University Bundang Hospital (https://www.snubh.org) Research Fund (No. 06-2018-140). The funders did not play a role in the study design, data collection and analysis, decision to publish, or preparation of the manuscript."

Reviewers' comments:

Reviewer's Responses to Questions

**Comments to the Author**

1. Is the manuscript technically sound, and do the data support the conclusions?

Reviewer #1: Partly

Reviewer #2: Yes

Reviewer #3: Yes

2. Has the statistical analysis been performed appropriately and rigorously? 

Reviewer #1: Yes

Reviewer #2: Yes

Reviewer #3: Yes

3. Have the authors made all data underlying the findings in their manuscript fully available?

Reviewer #1: No

Reviewer #2: Yes

Reviewer #3: Yes

4. Is the manuscript presented in an intelligible fashion and written in standard English?

Reviewer #1: Yes

Reviewer #2: Yes

Reviewer #3: Yes

5. Review Comments to the Author

Reviewer #1: Trends in the incidence and prevalence of dysphagia requiring medical attention among adults in South Korea, 2006–2016: A nationwide population study

General considerations:

- I suggest including in the first paragraph of your introduction the definition of dysphagia. It is not described anywhere. Here you can cite this paper: Castejón-Hernández, S., Latorre-Vallbona, N., Molist-Brunet, N., Cubí-Montanyà, D., & Espaulella-Panicot, J. (2020). Association between anticholinergic burden and oropharyngeal dysphagia among hospitalized older adults. Aging Clinical and Experimental Research, 33(7), 1981–1985. doi:10.1007/s40520-020-01707-9

- Please, do not use elderly to describe a population of older people. It is an ageist term. I suggest try to avoid language that might be deemed unacceptable or inappropriate (e.g. 'older people' is preferred to 'the elderly', the word 'senile' is best avoided). Take care with wording that might cause offence to ethnic or cultural groups”

- Try to avoid repetitive terms as dysphagia in every paragraph. Maybe you can describe your population as dysphagic or non-dysphagic. Even, you can use pronouns to refer to dysphagia in the same sentence.

- Maybe, it could be more interesting to describe whole/general prevalence more than a national one, because although Korea is a populous country, not all the countries have millions of population and these data could not be extrapolated into another countries.

- I am a little confused with the main aim of this study so maybe you should me more exact. If your objective is to know the prevalence of dysphagia independently of the cause (including mechanical, oropharyngeal, neurological … causes), please be explicit. In my opinion, it is not so relevant to determinate if they require medical attention or not. You only want to know a rate: the prevalence (percentage) of people affected by dysphagia

- I am not sure about if the insertion of nasogastric tube insertions should be considered as diagnostic criteria of dysphagia. What about surgical procedures? Or abdominal acute pain? Constipation?... They are clearly not related to dysphagia.

- I understand neurological disorders and cancer as a causing dysphagia… but COPD maybe is not the most important/prevalent cause of it… Please, check this bibliography which seems to be more philosophic than scientific (the journal has not impact factor).

Specific considerations:

Discussion:

- I propose you to add at the end of line n. 211: “and also some side-effects from sedative drugs, opioids and also higher anticholinergic burden”. Here you can cite Castejon et al.

- I suggest adding after “management” in line 228: “independently of the underlying diagnosis, because in older adults the most prevalent type of dysphagia is oropharyngeal cause”.

- In line 252, after “of dysphagia cases” it could be added the following sentence: “even the mechanical cause of dysphagia is not the most common/prevalent in older people”.

- If you cite the paper I recommended you before, you can include in line 266: hospital/institution setting is “ranged from 25.5% to 55%”.

- Consider to add at the end of line 289: “including the review of potentially causing medications or their side effects, as sedation or xerostomia”.

- Lines 306 and 307: Maybe they should be removed because of I explained before about the nasogastric tube.

- Lines 343 and 343: I propose to substitute “As they are common geriatric diseases and” for “As many countries are entering an aging or super-aging society, dysphagia will be a more common problem that requires appropriate interventions in daily clinical practice such as the rest of geriatrics syndromes”.

Thanks for your consideration! Great job but you need to continue working on your paper! Good luck!

Reviewer #2: This article seems to represent a unique review of dysphagia based on the intended population.

The article is easy to follow in its entirety.

Several considerations for review/minor editing:

Line 22-24: perhaps these sentences can be combined: ex "The prevalence of dysphagia and the resulting socioeconomic burden are known to be increasing, here we..."

Line 41-42: "elderly group older than 60 years" can probably be replaced with "geriatric population" as geriatric is a term used more frequently in the paper overall

Line 58: Perhaps "the cause of" should be "associated with" as there isn't necessarily a causal relationship demonstrated.

Line 59: "or" should be "and" ; "Indeed" is not needed.

Lines 61-65: Reviewing the sources here, would caution using statistics from other countries given the paper here is specifically a South Korean analysis.

Line 66: "However" is not needed

Line 73: Consider adding "the" after of

Line 74: Perhaps instead of "in a large, representative," can simply state "from a". It is assumed a population database has those characteristics.

Line 78: "Therefore" not needed

Line 80: "Thus" not needed

Line 87: consider "universal" instead of "mandatory"; at least in the United States the term "universal healthcare" is used, I can't comment on Canada or Europe.

Line 89: consider adding an ambiguous word such as "presumptively" after NHIS; this article makes a point of highlighting the weakness of the reporting system in the discussion. It is doubtfully a perfect system.

Line 100 (the entire section): The parameters investigated were inclusive and well described.

Lines 112-117: Would use the same formatting used earlier in the section: replace the period after dysphagia in line 113 with a colon (:), then use commas (,) instead of semicolons (;) before "2)" and "3)" and "and 4)".

Lines 113-114 - The context of the sentence sounds like something was edited out or was intended to be removed but was kept. Were there also codes used for specific brain imaging to identify stroke that were supposed to be given ICD-10 codes?

Line 116 - perhaps in this section "cancer" should be described more. It was in fact elaborated on in the discussion, but it might be more consistent to given an explanation here, as this is the definitions section of the article.

Line 119 - "many" is not needed

Line 122 - the first comma before "including" is not needed

Line 130 - perhaps instead of "from those at risk of developing dysphagia newly" could state something like "as they would not represent new/novel cases".

Line 131 - consider adding "dysphagia" after "incidence of" ; "The age-adjusted incidence and the incidence of [dysphagia in] each age group..."

Line 136 - again consider adding "dysphagia" to specify ; "The age-adjusted prevalence and prevalence [of dysphagia in] each age group was also calculated."

Line 138 and 144- I didn't catch why 2010 was written here when the title mentions data from 2006 to 2016.

Line 147 - consider "a" instead of "the" when referring the Kaplan-Meier curve here.

Fig. 1 and 2 - Consider including the labels mentioned in lines 158-159 and 167-168 on the actual figure to simplify interpretation of the graphs

Line 175- Consider "higher" instead of "high"

Line 181 - wording of the introductory sentence can be modified. Perhaps something like the following would make the point more easily understood: "The risk of dysphagia among the sampled comorbidities is noted from greatest to less association as follows:"

Line 182 - delete "s" in intervals

Lines 187 - 189 - it makes sense how it is written but not the easiest to follow. Perhaps something like "The Kaplan-Meier plot presented in Fig. 3 demonstrates a positive correlation between dysphagia and the risk of mortality. In addition, it is shown that this mortality risk increases yearly, particularly after the first 2 years of diagnosis."

Line 200 - consider adding something like "it is shown that" after the first comma

Line 201-202 - perhaps consider simply stating "geriatric population" rather than numbers (60 and 70 were both used, from my review of the charts the association with the >70 age range is quite obvious but the 60s, although is perhaps statistically significant based on numbers, is not visually as obvious)

Line 221 - Consider a period after 2019, and then start a next sentence starting with Cardiovascular (delete "and that")

Line 223 - Consider "demonstrates" instead of "demonstrated"

Line 235 - consider adding "possibly" after the common following "deficits". It is known that stroke is a risk factor for dysphagia but not all strokes cause dysphagia. The essence of this point was described well in the text and does not need elaboration.

Line 264 and 265 - "that" are not needed

Line 279 - "community dwellings" - I'm assuming this is a reference to the reader in regards to the nature/method of the prior studies cited earlier in the article. Perhaps here this can be reminded to the reader (this was line 204-206). I don't think this has be be a change per-se. I do believe lines 204-206 are well placed and should remain early in the discussion.

I don't have any specific comments on the remainder of the article other than it highlighted the strengths and in particular the weaknesses and limitations of the article very well. Follow-up research could certainly be done based on the example this article has set.

The conclusion is well stated.

The references section was not reviewed by me.

The supporting figures and tables were reviewed and were easy to interpret. My only critique would be adding labels to Figures 1 and 2 as stated above.

Reviewer #3: Kwon etal have studied the prevalence and incidence of dysphagia requiring medical attention among the adult Korean population available from NHIS database which covers 97% of Korean population.Using a stringent inclusion criteria of dysphagia -they found a lower prevalence of incidence and prevalence of the condition in their adult population,but increasing over the years,with significant increase over 70 years of age.A good study with appropriate methodology for detecting the nation wide trends in dysphagia,its predictors and outcome,these are my comments.

1.The operational definition of dysphagia used requires 2 assessments in 1 month or 2 instumental evaluation in 3 months.This could have excluded subjects with mild -moderate dysphagia who would have recovered also.Literature suggests that other than neurodegenerative disorders,others like stroke and cancer can be associated with transient rather than chronic dysphagia which also impacts short and long term outcome.Have they looked into this aspect while designing the study?

2.This exclusion of subjects with transient or short term dysphagia must have been the reason for low incidence and prevalence in their study cohort.

3.Have they included dysphagia assessment in only hospital settings or whether family physician assessment was also taken into account? Elderly with Presbydysphagia seems not to be included in this study cohort.

6. PLOS authors have the option to publish the peer review history of their article (what does this mean?). If published, this will include your full peer review and any attached files.

Reviewer #1: No

Reviewer #2: No

Reviewer #3: **Yes: **Sapna Erat Sreedharan

---

## [Author Response · Author response to Decision Letter 0]

12 Mar 2023

We greatly appreciate the efforts of the reviewers for their constructive comments in evaluating our manuscript. Each comment has been carefully considered and responses have been provided for each point.

---

## [Decision Letter · Decision Letter 1]

11 Apr 2023

PONE-D-22-35490R1Trends in the incidence and prevalence of dysphagia requiring medical attention among adults in South Korea, 2006–2016: A nationwide population studyPLOS ONE

Dear Dr. Kim,

Thank you for submitting your manuscript to PLOS ONE. After careful consideration, we feel that it has merit but does not fully meet PLOS ONE’s publication criteria as it currently stands. Therefore, we invite you to submit a revised version of the manuscript that addresses the points raised during the review process.

The authors need to amend the remaining minor yet important comments of the two of the reviewers to finalize the draft.

We look forward to receiving your revised manuscript.

Kind regards,

Sina Azadnajafabad

Academic Editor

PLOS ONE

Journal Requirements:

Reviewers' comments:

Reviewer's Responses to Questions

**Comments to the Author**

1. If the authors have adequately addressed your comments raised in a previous round of review and you feel that this manuscript is now acceptable for publication, you may indicate that here to bypass the “Comments to the Author” section, enter your conflict of interest statement in the “Confidential to Editor” section, and submit your "Accept" recommendation.

Reviewer #1: All comments have been addressed

Reviewer #2: All comments have been addressed

Reviewer #3: All comments have been addressed

2. Is the manuscript technically sound, and do the data support the conclusions?

Reviewer #1: (No Response)

Reviewer #2: Yes

Reviewer #3: Yes

3. Has the statistical analysis been performed appropriately and rigorously? 

Reviewer #1: Yes

Reviewer #2: N/A

Reviewer #3: Yes

4. Have the authors made all data underlying the findings in their manuscript fully available?

Reviewer #1: Yes

Reviewer #2: Yes

Reviewer #3: Yes

5. Is the manuscript presented in an intelligible fashion and written in standard English?

Reviewer #1: Yes

Reviewer #2: Yes

Reviewer #3: Yes

6. Review Comments to the Author

Reviewer #1: Congratulations! You've made a Great Job! Your paper has improved substantially and now is ready to be published! You only should make some little changes. Please, read the document attached and everything will be alright!

Reviewer #2: The article, as was its first draft, was easy to read throughout. The revisions made have made it even easier to read. Adding labeling to the figures helped with the clarity and ease of interpretation. Appropriate revisions have been made in response to the points made by the other prior reviewers as well.

With minor stylistic revisions this article should be accepted.

The line numbers of the comments below are specifically in reference to the MARKED UP version of the re-submission.

A remaining clarification that needs to be made:

-Line 143 to 150 - It still does not make sense why there is a reference to the year 2010. The title would seem to indicate data would have been taken from 2006 to 2016. Why wasn't 2006 written in lines 145 and 150 instead of 2010? On line 96 it states that data was taken from 2006 to 2016. Perhaps a simple comment to clarify this for the reader should be added. I don't think most readers would notice this issue. Refer to comment 16 to reviewer 2 from the original submission.

Consider the following stylistic additions and deletions:

-Line 22 - consider "limited" instead of "specific" and add "s" after population ("populationS")

-Line 40 to 42 - consider deleting the sentence starting with "The presence of stroke...". This is repetitive to what is already mentioned in lines 34-37

-Line 46-47 - consider deleting this sentence it is repetitive/doesn't add anything new

-Line 51 - consider adding "," after [4]

-Line 56 - consider changing "with the aging of the population" to "with an aging population" - this would sound more in line with the rest of the article

-Line 215 - consider adding "and comorbidities" after "geriatric syndromes".

-Line 215 - 218 - the sentence following "[7-8]" doesn't really contribute the the text, but doesn't hurt it either. I would consider deleting it to shorten the discussion.

-Line 238 - Consider adding "," after "because"

-Line241 - Consider changing "were" to "are"

-Line 247 - Consider moving "possibly" after the ","

-Line 256- Consider changing "depend" to "depends"

-Line 261- Consider splitting the sentence into two, the most logical way would be to add a "." after "food passage", delete "and" and start the next sentence with "Treatment modalities..."

-Line 265- Consider ending the sentence with "." after "dysphagia cases" and deleting the rest of the sentence. I'm not sure what was trying to be said, but it likely was not necessary or adding anything important to the article.

-Line 276- Consider removing "our study showed that" to reduce repetition

-Line 305- Consider adding "," after "medical attention"

-Line 317- Consider changing "the instrumental" to "an instrumental" and making tests singular (tests -> test).

-Line 324- Consider changing "dysphagia in the previous" to "dysphagia in a previous"

-Line 324- Consider adding something like ", and we utilized this as well." after Taiwan.

-Line 359- Consider changing "seen in aging population in developed" to "seen in the aging populations of other"

-Line 360- Consider ending the sentence after "countries" and deleting the rest of the sentence.

Reviewer #3: Authors have gone through the reviewers comments carefully and have addressed them well,adding to the quality of manuscript.

7. PLOS authors have the option to publish the peer review history of their article (what does this mean?). If published, this will include your full peer review and any attached files.

Reviewer #1: No

Reviewer #2: No

Reviewer #3: **Yes: **SAPNA ERAT SREEDHARAN

---

## [Author Response · Author response to Decision Letter 1]

24 May 2023

Thank you for the constructive comments.

---

## [Decision Letter · Decision Letter 2]

7 Jun 2023

Trends in the incidence and prevalence of dysphagia requiring medical attention among adults in South Korea, 2006–2016: A nationwide population study

PONE-D-22-35490R2

Dear Dr. Kim,

We’re pleased to inform you that your manuscript has been judged scientifically suitable for publication and will be formally accepted for publication once it meets all outstanding technical requirements.

Kind regards,

Sina Azadnajafabad, MD, MPH

Academic Editor

PLOS ONE

Additional Editor Comments (optional):

Reviewers' comments:

Reviewer's Responses to Questions

**Comments to the Author**

1. If the authors have adequately addressed your comments raised in a previous round of review and you feel that this manuscript is now acceptable for publication, you may indicate that here to bypass the “Comments to the Author” section, enter your conflict of interest statement in the “Confidential to Editor” section, and submit your "Accept" recommendation.

Reviewer #1: All comments have been addressed

Reviewer #2: All comments have been addressed

2. Is the manuscript technically sound, and do the data support the conclusions?

Reviewer #1: Yes

Reviewer #2: Yes

3. Has the statistical analysis been performed appropriately and rigorously? 

Reviewer #1: Yes

Reviewer #2: N/A

4. Have the authors made all data underlying the findings in their manuscript fully available?

Reviewer #1: Yes

Reviewer #2: Yes

5. Is the manuscript presented in an intelligible fashion and written in standard English?

Reviewer #1: Yes

Reviewer #2: Yes

6. Review Comments to the Author

Reviewer #1: Congratulations! You made a great job! You have understood and incorporated all the suggestions made by the reviewers.

Reviewer #2: The previous queries have all been addressed satisfactorily.

The publication is easy to read through, the primary points of interest clearly and thoroughly reviewed.

7. PLOS authors have the option to publish the peer review history of their article (what does this mean?). If published, this will include your full peer review and any attached files.

Reviewer #1: No

Reviewer #2: No

---

## [Editor Report · Acceptance letter]

20 Jun 2023

PONE-D-22-35490R2 

Trends in the incidence and prevalence of dysphagia requiring medical attention among adults in South Korea, 2006–2016: A nationwide population study 

Dear Dr. Kim:

I'm pleased to inform you that your manuscript has been deemed suitable for publication in PLOS ONE. Congratulations! Your manuscript is now with our production department. 

Kind regards, 

on behalf of

Dr. Sina Azadnajafabad 

Academic Editor

PLOS ONE